# Multi-scale dynamics influence the division potential of stomatal lineage ground cells in *Arabidopsis*

Hannah F. Fung [1,2], Gabriel O. Amador [3], Renee Dale [4], Yan Gong [1,5], Macy Vollbrecht [1], Joel M. Erberich [1], Andrea Mair [2] & Dominique C. Bergmann [1,2] ✉

During development, many precursor lineages are flexible, producing variable numbers and types of progeny cells. What determines whether precursors differentiate or continue dividing? Here we take a quantitative approach that combines long-term live imaging, statistical modeling and computational simulations to probe the developmental flexibility of stomatal lineage ground cells (SLGC) in *Arabidopsis* leaves. We discover that cell size is a strong predictor of SLGC behaviour and that cell size is linked to division behaviour at multiple spatial scales. At the neighbourhood scale, cell size correlates with the strength of cell-cell signaling, which affects the rate at which SPEECHLESS (SPCH), a division-promoting transcription factor, is degraded. At the subcellular scale, cell size correlates with nuclear size, which modulates the concentration of SPCH in the nucleus. Our work shows how initial differences in SPCH levels are canalized by nuclear size and cell-cell signaling to inform the behaviour of a flexible cell type.

During development, different precursor lineages give rise to the full complement of cell types in a multicellular organism. Some lineages are more rigid, generating a fixed number of progeny of a certain type, while others are more flexible, producing variable numbers and types of progeny. The latter includes many plant lineages, which respond to environmental conditions to build tissues of different sizes and composition throughout an organism's life. The *Arabidopsis* stomatal lineage offers a tractable system in which to investigate the emergence of stereotyped, but flexible patterns.

In the developing leaf, stomatal lineage cells undergo a series of asymmetric cell divisions (ACDs) that produce two daughter cells. The smaller daughter, called the meristemoid, can either differentiate into a guard mother cell (GMC) and ultimately a stoma, or divide asymmetrically one to five times before differentiating (Fig. 1a). The larger daughter, called the stomatal lineage ground cell (SLGC), faces a similar choice: it can either differentiate into a pavement cell or divide asymmetrically to generate a meristemoid and SLGC (Fig. 1a). SLGCs

are often described as the larger "differentiating daughters", destined to form pavement cells[1-3], but this is a mischaracterization. SLGCs do divide asymmetrically, though at lower frequencies than meristemoids[4].

A growing body of work suggests that SLGC divisions are actively suppressed. During asymmetric cell divisions, the mother cell segregates several polarity proteins to the SLGC, where they act as molecular scaffolds for a mitogen-activated protein kinase (MAPK) signaling cascade that promotes the degradation of SPEECHLESS (SPCH), a transcription factor required for asymmetric cell divisions[2,5-9]. In contrast, SPCH levels remain high in the meristemoid sister, allowing it to divide multiple times before differentiating.

Although this model explains why meristemoids divide more often than SLGCs, it does not account for the flexibility of SLGC behaviours: why do some SLGCs divide, while others differentiate? This gap in knowledge is striking, given SLGCs have profound effects on the cell type composition of a leaf, responding to external inputs to

[1]Department of Biology, Stanford University, Stanford, CA 94305, USA. [2]Howard Hughes Medical Institute, Stanford, CA 94305, USA. [3]Department of Developmental Biology, Stanford University, Stanford, CA 94305, USA. [4]Donald Danforth Plant Science Center, St. Louis, MO 63132, USA. [5]Department of Organismic and Evolutionary Biology, Harvard University, Cambridge, MA 02138, USA. ✉e-mail: dbergmann@stanford.edu

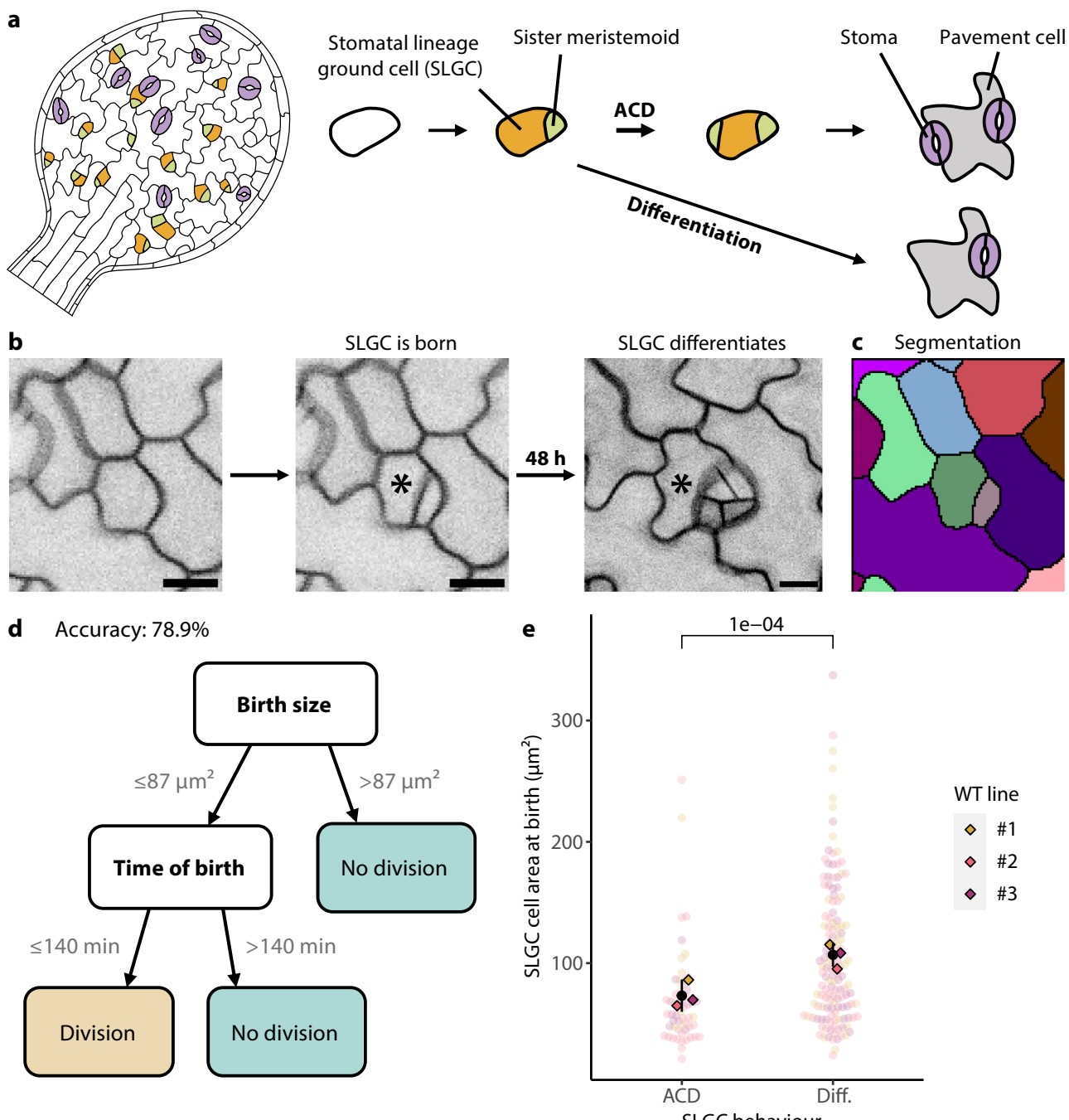

**Fig. 1 | A decision tree identifies birth size as the strongest predictor of SLGC behaviour. a** A cartoon of a developing *Arabidopsis* cotyledon (left), with a diagram of the stomatal lineage (right). Stomatal lineage cells divide asymmetrically to produce a smaller meristemoid (green) and a larger stomatal lineage ground cell (SLGC, gold). Meristemoids ultimately differentiate into stomata (purple). SLGCs can either divide asymmetrically (ACD) or differentiate into pavement cells (grey). **b** Illustration of the imaging-based approach. For each newly born SLGC (marked with an asterisk), we measured 15 features at birth. Two days later, we re-imaged the cell to capture its behaviour. In this example, the SLGC differentiated. Scale bar: 10 μm. **c** Cell segmentation of the second time point in (**b**), for semi-automated quantification of cellular features. **d** Decision tree following cost complexity pruning. Birth size was the strongest predictor of SLGC behaviour, followed by time of birth. **e** Cell areas at birth of SLGCs that divided (ACD) or differentiated (Diff.). Black circles and lines are individual-level means and standard deviations. The *P*-value is from a mixed-effects model with behaviour as a fixed effect and individual as a random effect. *N* = 3 individuals; 50, 62, 80 cells per individual. Source data and statistical details are provided in Source Data file 1.

generate three-quarters of all stomata[10,11]. What are the factors that predispose an SLGC toward division or differentiation? Do the behaviours of individual SLGCs reflect their lineage history or their cellular neighbourhood?

Past efforts to characterize SLGCs have been hampered by the lack of cell-type-specific markers that distinguish SLGCs from their sister meristemoids in unequivocal ways. Consequently, it has been challenging to isolate SLGCs for transcriptomics[12] or to employ

molecular techniques that rely on cell-type-specific promoters to manipulate cells. The subtle and quantitative phenotypes expected from the loss of regulators of SLGC behaviour make forward genetic screens infeasible.

Over the past two decades, quantitative studies have been instrumental in driving our understanding of processes that have eluded more traditional genetic approaches. For example, in the *Drosophila* embryo, careful in vivo measurements of the Bicoid transcription factor have offered fresh insight into the mechanisms by which morphogen gradients are established[13]. Quantitative analyses have also overturned models: a recent study of the *Arabidopsis* root challenged a model of how formative cell divisions are regulated in the stem cell niche[14].

Here we developed a quantitative approach that combines long-term imaging and statistical modeling to identify correlates of SLGC behaviour at multiple spatiotemporal scales. Using this approach, we discovered that cell size is a strong predictor of SLGC behaviour: larger SLGCs divide less often than smaller cells. While we recently reported a size-based fate decision for other leaf epidermal cells[15], in this study we go further by providing a molecular explanation for the link between SLGC size and behaviour. We propose that cell size is linked to division behaviour at multiple spatial scales. At the neighbourhood scale, cell size correlates with the strength of cell-cell signaling, which affects the rate at which SPCH is degraded. At the subcellular scale, cell size correlates with nuclear size, which modulates the concentration of SPCH in the nucleus. Our work shows how initial differences in SPCH levels are canalized by nuclear size and cell-cell signaling to inform the behaviour of a flexible cell type.

## Results

### Birth size emerges as the strongest predictor of SLGC behaviour

Given the importance of SLGCs in leaf flexibility, but the challenges in using traditional genetic approaches to identify factors regulating this cell type, we adopted a holistic, imaging-based approach. We developed an imaging pipeline to measure cellular features at multiple scales, capturing SLGCs in an intact, wild-type, 3-day post germination (dpg) cotyledon from birth to their final, developmental outcomes (Fig. 1b). This 3-dpg timepoint was chosen to maximize the number of active stomatal lineage cells for subsequent tracking. We then segmented the cell outlines using ilastik[16], which enabled the semi-automated quantification of 15 features at birth, including time of birth, tissue position, cell size and shape, mother identity, and characteristics of the immediate neighbours (Supplementary Fig. 1a). Two days later, we re-imaged the cotyledon and recorded whether each SLGC divided or differentiated (Fig. 1b). This two-day interval captures the vast majority of SLGC divisions (Supplementary Fig. 1b). Cells in the cotyledon epidermis are of nearly uniform height (Z-dimension) at these stages[17,18] and for efficiency of analysis, we tracked areal growth, as is common in the field[19].

To pinpoint the most predictive features, we fed our measurements into a classification and regression tree (decision tree), which selects predictive features and orders them from most to least predictive. After cost complexity pruning to prevent overfitting (Supplementary Fig. 1c), we obtained a decision tree with a prediction accuracy of 78.9% (Fig. 1d). SLGC birth size was the strongest predictor, followed by time of birth (Fig. 1d). We also specified a random forest classifier which yielded very similar results: birth size had the highest feature importance score, followed by time of birth (Supplementary Fig. 1d).

To ensure these findings were robust, we captured and analyzed time-lapses of two other wild-type lines bearing different fluorescent reporters (see Methods). Across the three lines, size was consistently predictive of behaviour: only smaller cells could divide again, while larger cells differentiated (Fig. 1e). Surprisingly, this is the opposite of what we see in meristemoids, where the probability of dividing asymmetrically increases with cell size[15].

### Larger cells are born with lower SPCH concentrations

To understand why the relationship between size and behaviour is inverted in SLGCs (relative to meristemoids), it is useful to identify the specific genes or proteins involved. We therefore turned our attention to one of the few well-characterized proteins present in SLGCs, the transcription factor SPCH. Previous work reported that the frequency of SLGC divisions increased upon cytokinin signalling manipulations, and that the SLGCs expressed SPCH before dividing[11]. Whether SPCH has a similar role during normal development, however, was not determined. Nevertheless, the cytokinin results provide testable hypotheses about the relationship between SPCH levels and size-dependent divisions, namely that (1) SPCH levels should correlate with SLGC behaviour; and (2) larger cells should contain less SPCH.

To quantify SPCH levels, we captured time-lapses of 3-dpg cotyledons expressing a translational reporter (*pSPCH::SPCH-YFP* rescuing *spch-3-/-*)[15,20] and tracked SPCH intensities from cell birth to the end of the time-lapse (Fig. 2a, b). SPCH was exclusively nuclear during this period. Surprisingly, SPCH intensities were already predictive at birth: dividing cells were born with significantly more SPCH than differentiating cells (Fig. 2c).

After birth, SPCH intensities declined dramatically for ~200 min in both dividing and differentiating populations (Fig. 2d). While repeated imaging can lead to photobleaching and a decline in SPCH intensities, we confirmed that bleaching alone could not account for a decline of this magnitude (see Methods, Supplementary Table 1). At ~200 minutes after birth, SPCH levels began to rise in dividing, but not in differentiating cells (Fig. 2d). We subsequently refer to this nadir in SPCH intensity as the "dip".

To determine whether the rise in SPCH levels was a consequence of cell cycle progression, we compared SPCH dynamics to that of the replication licensing factor CDT1A, which accumulates during G1 and is rapidly degraded at the G1/S transition[21]. The rise in SPCH intensities preceded that of CDT1A by ~100 minutes (Supplementary Fig. 2a), implying that SPCH is a cause, rather than a consequence of cell cycle progression. The dynamics of SPCH nuclear concentration over time (SPCH intensity divided by nuclear area) resembled those of SPCH intensity (Fig. 2e).

Next, we tested whether larger SLGCs contain less SPCH. There are two ways of measuring the 'amount' of SPCH in a cell: fluorescence intensity, which scales with the number of molecules of SPCH, and nuclear concentration, which also accounts for nuclear size. Of the two measures, concentration is more biologically meaningful because it contributes directly to transcription by affecting binding site occupancy[22]. Although we did not detect a significant correlation between SPCH intensity and cell size at birth (Fig. 2f), we found that larger cells were born with lower nuclear concentrations of SPCH (Fig. 2g). Since larger cells have larger nuclei (Supplementary Fig. 2b), these data suggest that SPCH is diluted in larger cells (Fig. 2h). Taken together, our results indicate that larger cells are born with lower concentrations of SPCH, which may explain why they divide less often.

### Large SLGCs divide more often when SPCH levels are increased

A SPCH "dilution" hypothesis predicts that large cells will divide more often if their SPCH concentration is increased. To address this prediction, we first determined whether large cells are capable of responding to a general division-promoting factor. We expressed the D-type cyclin CYCLIN D7;1 (CYCD7;1) under the epidermis-specific *ATML1* promoter (*pATML1::CYCD7;1-YFP*)[23] in wild-type cotyledons. The construct was expressed in all epidermal cells at 3-dpg (Supplementary Fig. 3a) and induced both small and large SLGCs to divide (Supplementary Fig. 3b, c). Thus, we can conclude that large SLGCs are division-competent.

Next, we tested whether large cells divide more often when supplied with more SPCH. We generated a *pSPCH::SPCH-YFP; spch-3* line where SPCH was overproduced in its normal expression domain

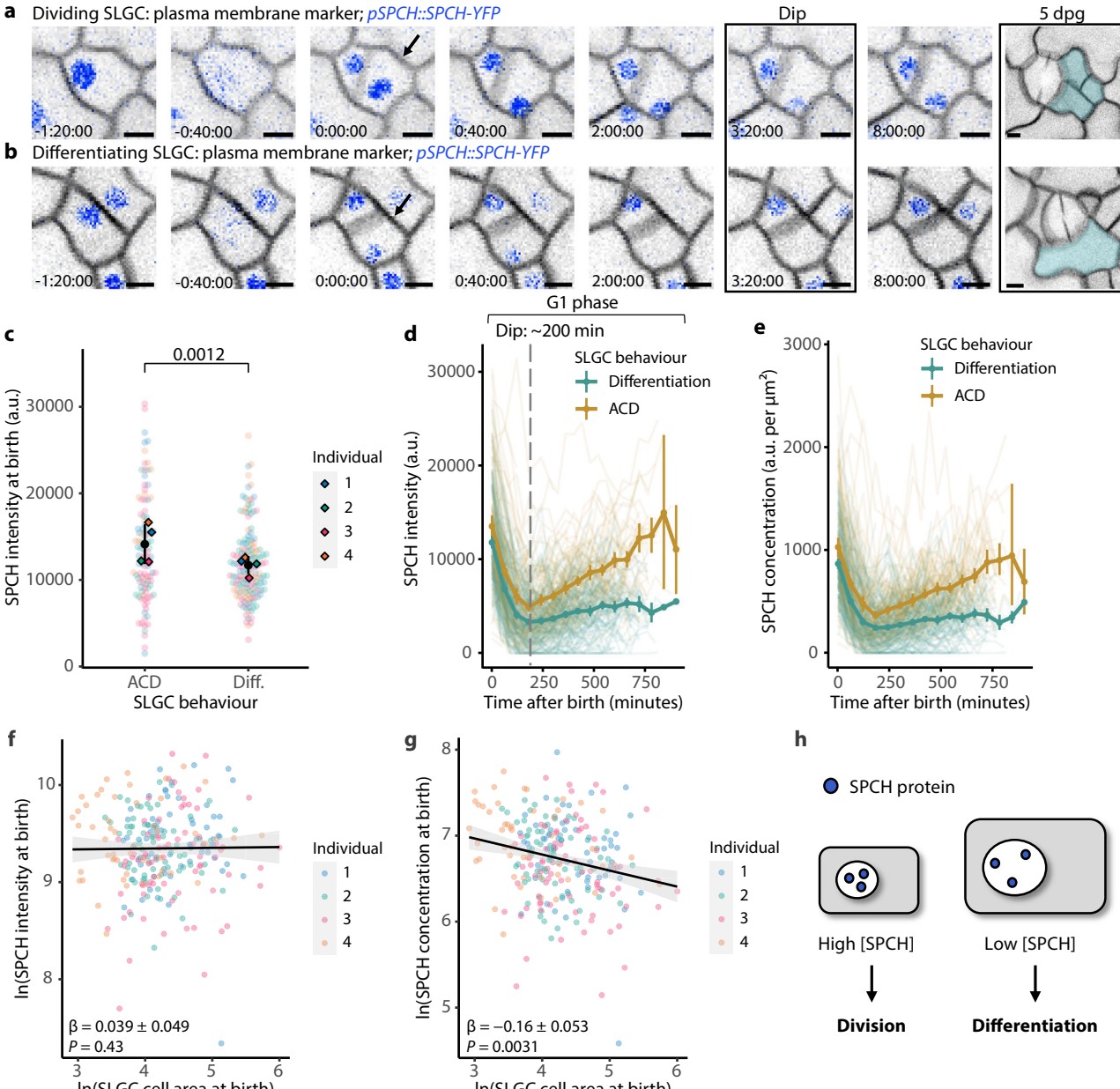

**Fig. 2 | Larger cells are born with lower concentrations of SPCH and divide less often.** Inverted confocal images of the SPCH translational reporter *pSPCH::SPCH-YFP; spch-3* in a dividing cell (**a**) or a differentiating cell (**b**) at 3-dpg. Final image is false colored blue to indicate the final fate of the tracked SLGC. The SLGCs were born at 0 min (arrows). Scale bar: 5 μm. **c** SPCH intensities at birth in dividing (ACD) or differentiating (Diff.) cells. Black circles and lines are individual-level means and standard deviations. SPCH intensities over time (**d**) or SPCH nuclear concentrations over time (**e**), coloured by behaviour. Circles and vertical lines are binned means and bootstrapped 95% confidence intervals. SPCH intensity at birth (**f**) or SPCH nuclear concentration at birth (**g**) vs. cell area at birth. Axes are ln-transformed. Black lines and grey bands are linear model predictions and 95% confidence intervals. **h** A cartoon of the SPCH dilution model. Small and large cells are born with comparable SPCH intensities, a proxy for the number of SPCH molecules. Because larger cells have larger nuclei, they are born with lower concentrations of SPCH and divide less often. **c**, **f**, **g** P-values are from mixed-effects models with individual as a random effect. **c**–**g** *N* = 4 individuals; 50, 62, 75, 75 cells per individual. Source data and statistical details are provided in Source Data file 1.

(SPCH++; see Methods). Across cell sizes, SPCH intensities were higher in SPCH++ cotyledons than in those expressing the SPCH translational reporter (Supplementary Fig. 3d). Accordingly, we observed an increase in the proportion of SLGCs that divided, relative to wild-type cotyledons (Fig. 3a). This increase in division frequency was due to elevated SPCH levels, rather than a significant change in birth size (Fig. 3b).

To determine specifically whether large cells divide more often when SPCH is overproduced, we binned wild-type and SPCH++ cells into quintiles based on size. For each quintile, we computed the proportion of cells that divided per genotype (Fig. 3c). Consistent with our hypothesis, SPCH++ cells divided more often than wild-type cells in the fourth and fifth quintiles, indicating that *SPCH* overexpression is sufficient to increase the proportion of large cells that divide (Fig. 3c).

### Number of signaling neighbours can influence SPCH degradation rates

One inescapable observation in the time-lapse imaging of SPCH is that this factor is highly dynamic. SPCH intensities decline after birth in all SLGCs (Fig. 2d), but interestingly, they appear to fall faster in

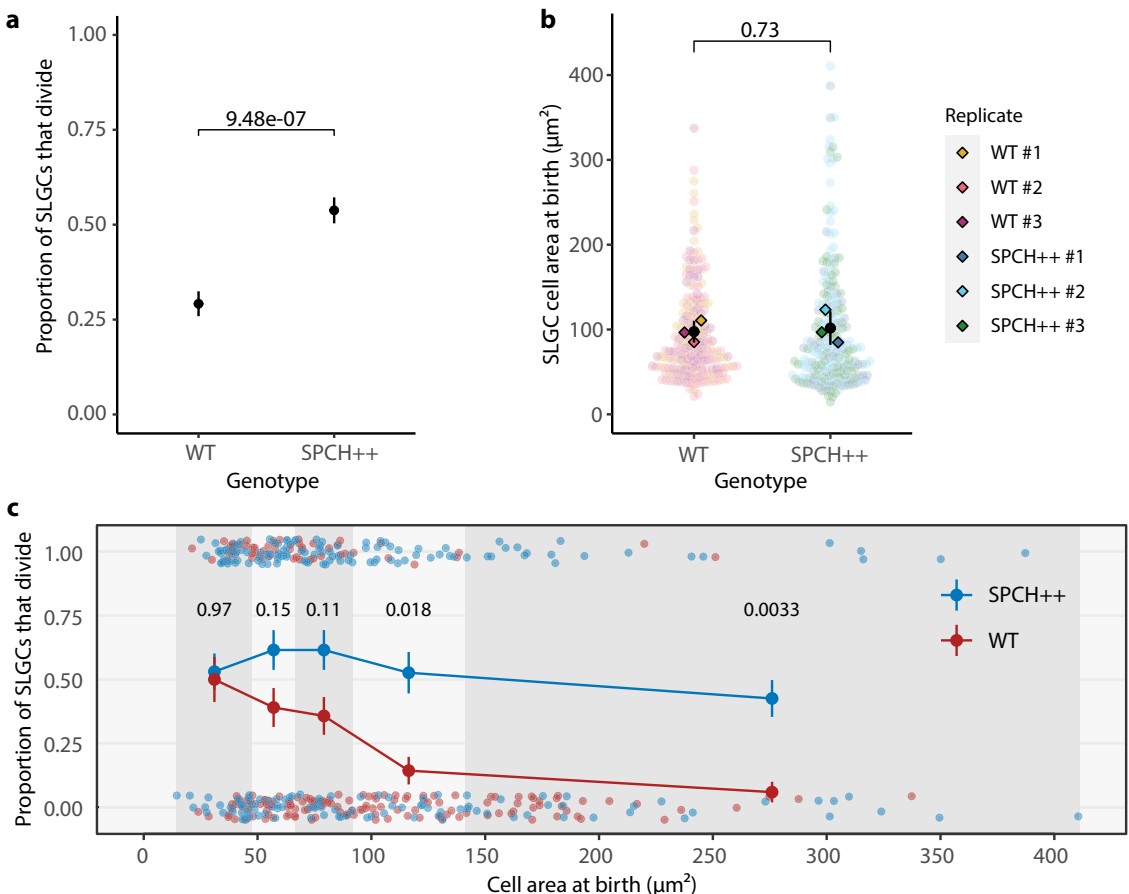

**Fig. 3 | Large SLGCs divide more often when SPCH levels are increased.** The proportion of SLGCs that divided (**a**) or SLGC cell areas at birth (**b**) in wild-type seedlings and seedlings where SPCH accumulates to higher levels (strong *pSPCH::SPCH-YFP spch-3*; SPCH + +). Black circles and lines are means and standard deviations. *P*-values are from a two-sided two-sample test for equality of proportions with continuity correction (**a**) or a mixed-effects model with individual as a random effect (**b**). **c** The proportion of SLGCs that divided in wild-type and SPCH + + cotyledons. Vertical shading delineates quintiles of SLGC cell area at birth, from left: smallest 20%, 20–40%, 40–60%, 60–80%, largest 20%. Circles and lines are means and standard deviations. *P*-values are from Holm-Bonferroni corrected two-sided two-sample tests for equality of proportions with continuity correction. **a**–**c** *N* = 3 individuals per genotype; wild-type: 50, 62, 80 cells per individual; SPCH + +: 62, 70, 80 cells per individual. Source data and statistical details are provided in Source Data file 1.

differentiating cells. This prompted us to test whether the rate of SPCH decline is correlated with SLGC behaviour. We fit an exponential decay function ($N(t) = N_0 * e^{\lambda t}$) to the SPCH intensities of each cell from 0 to ~200 min after birth and estimated the decay constant $\lambda$ (the more negative the constant, the higher the rate of decline). The decay constant was predictive of SLGC behaviour: SPCH levels fell more rapidly in differentiating cells than in dividing cells (Supplementary Fig. 4a). Among cells where SPCH levels declined, larger cells showed higher rates of decline than smaller cells (Supplementary Fig. 4b).

Why would SPCH levels decline faster in larger cells? Previous studies have shown that SPCH is regulated by the peptides EPIDERMAL PATTERNING FACTOR 1 and 2 (EPF1/2)[24,25], which activate a MAPK signaling cascade that targets the SPCH protein in neighbouring cells for degradation (Fig. 4a)[26]. EPF1 and EPF2 are reported to be secreted by meristemoids, GMCs, and young stomata to prevent their neighbours from developing into stomata[25,27–29]. This ensures that stomata are spaced apart, which optimizes stomatal function[30,31]. Although mobile peptides could act over large spatial scales, lineage tracing in stomatal signaling mutants[10] suggests that the signals that establish and maintain stomatal spacing are likely juxtacrine.

In light of the known EPF-MAPK signaling pathway, a plausible explanation for why larger SLGCs experience higher SPCH degradation rates is that they have more "signaling neighbours" (neighbours that are meristemoids, GMCs, or stomata; Fig. 4a). This geometric argument predicts that (1) cells with more signaling neighbours should be

larger; (2) they should divide less often; and (3) they should experience higher SPCH degradation rates.

To test these predictions, we analyzed the time-lapses of 3-dpg cotyledons expressing the SPCH translational reporter used in Fig. 2. Cells with more signaling neighbours were larger (Fig. 4b) and divided less often (Fig. 4c). The number of signaling neighbours appeared to be a good proxy for the strength of EPF signaling, as neither the total number of neighbours (both signaling and non-signaling) nor the fraction of the cell perimeter in contact with a signaling neighbour was predictive of SLGC behaviour, after accounting for the number of signaling neighbours (Supplementary Fig. 4c, d). In line with our third prediction, cells with more signaling neighbours experienced higher SPCH degradation rates (Fig. 4d, e').

These data conform to textbook "lateral inhibition" models in which mature stomata generate inhibitory fields to prevent the formation of adjacent stomata[32]. However, a closer look at our data separated by cell type reveals that meristemoids are largely responsible for the neighbour effect. Cells with more meristemoid neighbours experienced higher SPCH degradation rates (Fig. 4f–g'). They were also larger (Fig. 4h) and divided less often (Fig. 4i).

In contrast, neither the number of adjacent stomata nor the number of adjacent GMCs was significantly associated with SPCH degradation rates (Fig. 4j, Supplementary Fig. 4e, f). Very few of the SLGCs at 3-dpg had a stomatal neighbour, compromising our ability to estimate the mean SPCH degradation rate in this group of cells.

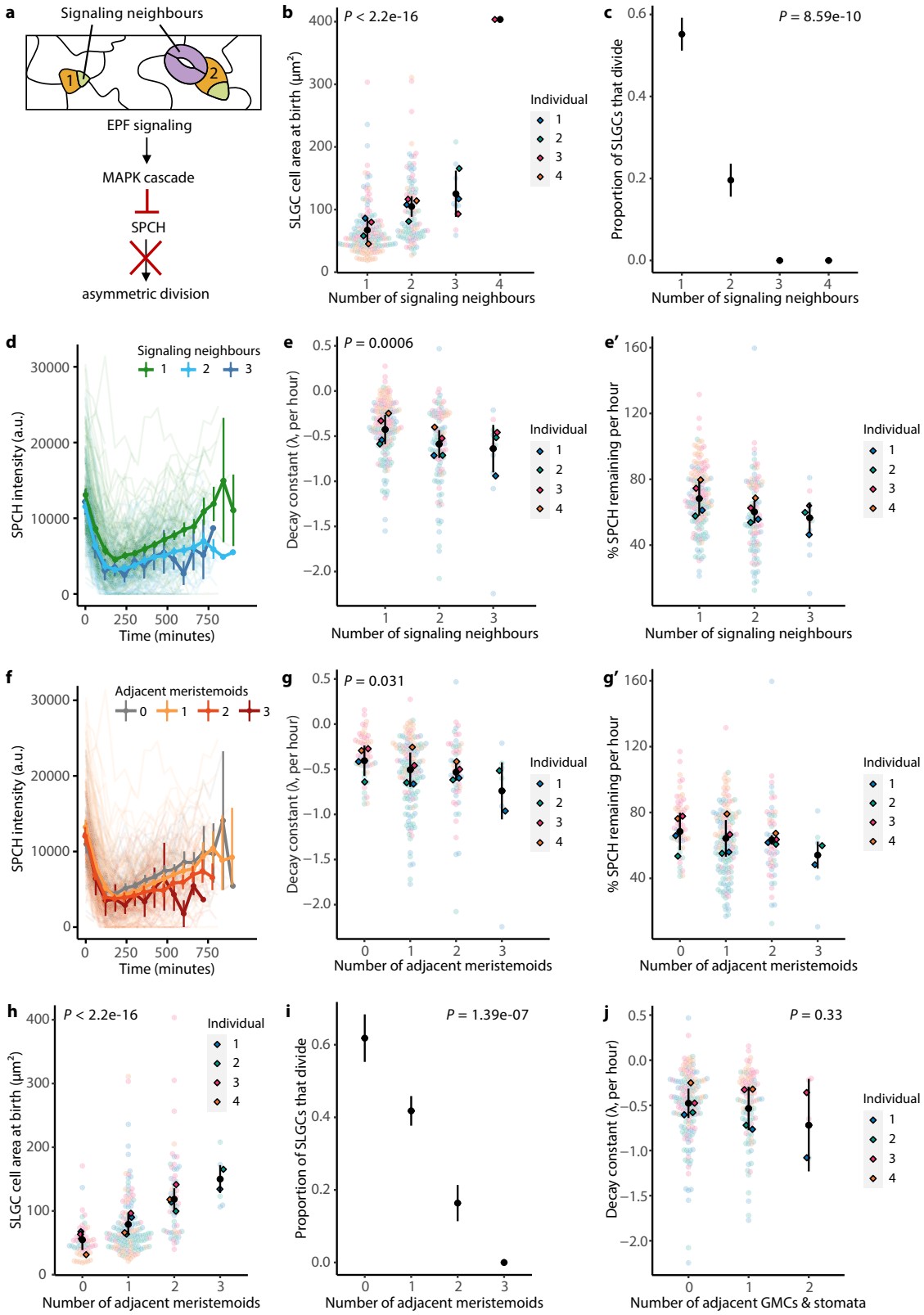

Consequently, our data lacked the statistical power to detect an association (if any) between the number of stomata and degradation rate. Our GMC results, on the other hand, were not limited by statistical power. Here we considered two explanations for the lack of GMC influence; first, that GMCs do not suppress SLGC divisions, or second, that GMCs suppress SLGC divisions independently of SPCH degradation. Contrary to expectations from lateral inhibition models, cells with

more GMC neighbours tended to divide more often (Supplementary Fig. 4h).

Finally, we tested whether the link between size and behaviour is abrogated when the ability of SPCH to respond to MAPK signaling is disrupted. In seedlings where *spch-3* is rescued with a SPCH variant lacking three MAPK phosphorylation sites (*pSPCH::SPCH2-4A-YFP*)[33], many large SLGCs (>150 μm²) divided (Fig. 5a). In fact, dividing cells

**Fig. 4 | Cells with more signaling neighbours experience higher SPCH degradation rates and divide less often. a** Cartoon of cells monitored in this figure (top), and diagram of the EPIDERMAL PATTERNING FACTOR (EPF) signaling cascade that targets SPCH for degradation (bottom). 1: SLGC with one signaling neighbour. 2: SLGC with two signaling neighbours. SLGC cell area at birth (**b**) or the proportion of SLGCs that divided (**c**) by the number of signaling neighbours. **d** SPCH intensities over time, coloured by the number of signaling neighbours. **e** The decay constant, a measure of how fast SPCH is degraded, by the number of signaling neighbours. The more negative the constant, the higher the degradation rate. **e′** The same data, re-plotted as the percentage of SPCH remaining after every hour. SPCH levels declined in most cells, but we did observe some cells where SPCH increased (points >100%). **f** SPCH intensities over time, coloured by the number of adjacent meristemoids. The decay constant (**g**), the percentage of SPCH remaining after every hour (**g′**), cell area at birth (**h**), or the proportion of SLGCs that divided (**i**) by the number of adjacent meristemoids. **j** The decay constant by the number of adjacent GMCs and stomata. **b**, **c**, **e**, **g**–**j** Black circles and lines are means and standard deviations. *P*-values are from mixed-effects models with individual as a random effect (**b**, **e**, **g**, **h**, **j**) or chi-squared tests for trend in proportions (**c**, **i**). **d**, **f** Circles and vertical lines are binned means and bootstrapped 95% confidence intervals. **b**–**j** *N* = 4 individuals; 50, 62, 75, 75 cells per individual. Source data and statistical details are provided in Source Data file 1.

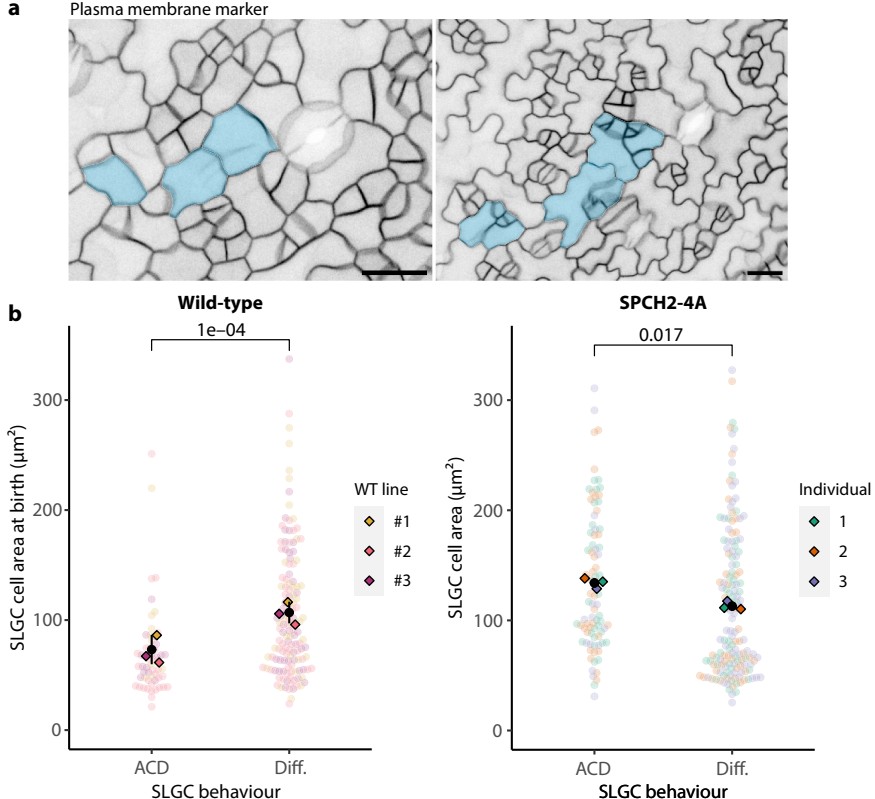

**a** Plasma membrane marker

**b**  Wild-type — 1e−04    SPCH2-4A — 0.017

SLGC cell area at birth (μm²) / SLGC cell area (μm²) by SLGC behaviour (ACD, Diff.)

WT line: #1, #2, #3 — Individual: 1, 2, 3

**Fig. 5 | The link between size and behaviour is abrogated when the ability of SPCH to respond to MAPK signaling is disrupted. a** Micrograph of a 3-dpg *spch-3* cotyledon expressing a SPCH variant lacking three MAPK phosphorylation sites (*pSPCH::SPCH2-4A-YFP*; left). The same region two days later (right). Large, dividing SLGCs are false-coloured in blue. Scale bar: 20 μm. **b** Cell areas of SLGCs that divided (ACD) or differentiated (Diff.) in wild-type and *pSPCH::SPCH2-4A-YFP; spch-3* (SPCH2-4A) cotyledons. The wild-type data are re-plotted from Fig. 1e. Black circles and lines are individual-level means and standard deviations. *P*-values are from mixed-effects models with behaviour as a fixed effect and individual as a random effect. *N* = 3 individuals per genotype; wild-type: 50, 62, 80 cells per individual; SPCH2-4A: 80 cells per individual. Source data and statistical details are provided in Source Data file 1.

were slightly but significantly larger than differentiating cells (Fig. 5b), consistent with MAPK signaling suppressing the division potential of large SLGCs.

## Simulations and statistical evaluations of models

So far, we have proposed that large cells divide less often because they experience higher rates of SPCH degradation and the SPCH they contain is diluted by larger nuclear compartments. In these cells, the concentration of SPCH ([SPCH]) remaining after degradation (at the "dip") is too low to activate the proposed positive feedback loop required to drive cell cycle progression[34,35]. Our model yields two predictions that can be tested statistically: [SPCH] at the dip should be predictive of SLGC behaviour; and among cells with the same SPCH intensities at the dip, cells with larger nuclei should divide less often.

In line with the first prediction, dividing cells had significantly higher [SPCH] at the dip than differentiating cells (Fig. 6a). Consistent with the second prediction, cells with larger nuclei divided

less often than those with smaller nuclei, after controlling for SPCH intensity at the dip (Fig. 6b). Both pathways converge on [SPCH] at the dip, which we hypothesize is a primary determinant of SLGC behaviour.

Is [SPCH] at the dip sufficient to recapitulate SLGC behaviours? To test this, we specified a stochastic and asynchronous rule-based lineage decision tree model, with [SPCH] at the dip as the sole determinant of behaviour (Fig. 7a, left). The model begins with a population of 1000 mother cells, each with a birth size, SPCH intensity, and number of signaling neighbours drawn from empirically derived distributions (see Supplementary Fig. 5 and Methods for details). The mother cells divide asymmetrically with a randomly drawn asymmetry parameter, each forming a smaller meristemoid and a larger SLGC (Fig. 7a, Supplementary Fig. 5). Based on the measurements in Supplementary Fig. 6, each SLGC is assumed to inherit two-thirds of its mother's SPCH intensity and to have one signaling neighbour more than its mother (i.e. the newly generated sister meristemoid).

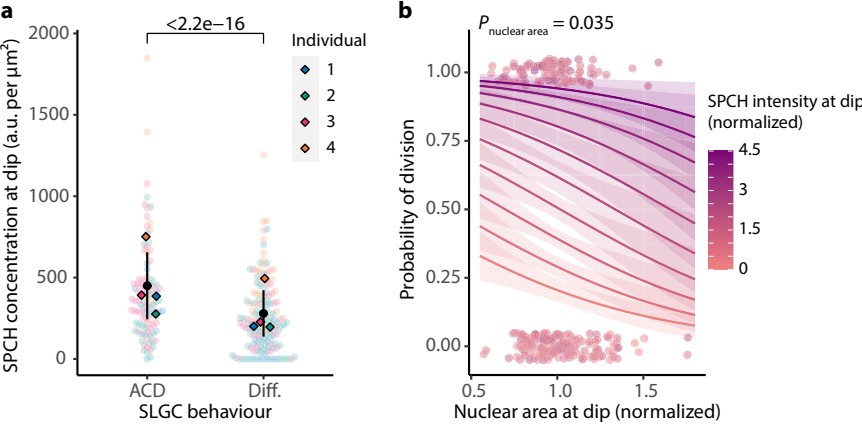

**Fig. 6 | [SPCH] at the dip is predictive of SLGC behaviour. a** SPCH nuclear concentrations at the dip in dividing (ACD) or differentiating (Diff.) cells. The *P*-value is from a mixed-effects model with behaviour as a fixed effect and individual as a random effect. **b** Multiple logistic regression of the probability of division on nuclear area at the dip, controlling for SPCH intensity at the dip. Among cells with the same SPCH intensity at the dip, cells with larger nuclei divided less often, because the SPCH they contained was diluted by larger nuclear compartments. Data from different individuals were normalized (relative to the individual mean) and pooled for visualization. Lines and bands are logistic model predictions and standard errors. The *P*-value is from a logistic regression with individual as a blocking variable. $N = 4$ individuals; 42, 61, 72, 75 cells per individual. Source data and statistical details are provided in Source Data file 1.

After birth, SPCH is degraded according to one of four modes: the decay constants ($\lambda$) are randomly drawn ("random"); modulated by signaling neighbours ("neighbour-based"); modulated by size ("size-based"); or modulated by both signaling neighbours and size ("neighbour- and size-based"; see Methods). The SLGCs then undergo a fate-determining program in which cells with higher [SPCH] have a higher chance of dividing (Fig. 7a, left). We derived all input parameters by fitting theoretical probability distribution functions or logistic regressions to empirical measurements from each of four individual plants (Supplementary Fig. 5; see Supplementary Tables 2 and 3 for fitted parameters).

We assessed each "SPCH-only" model according to its ability to recapitulate the sizes and [SPCH] of dividing and differentiating cells (Supplementary Fig. 7a, b), and the proportion of cells that divided given the number of signaling neighbours (Fig. 7b, Supplementary Table 4; see Methods for details). The model with the lowest error was one where degradation rates scaled with both signaling neighbours and size ("best [SPCH] only" model, Supplementary Fig. 7a, b). However, it struggled to reproduce the negative relationship between signaling neighbours and behaviour (Fig. 7b), suggesting that [SPCH] alone is insufficient to recapitulate the observed patterns.

To determine whether other features were required for fate determination, we specified fate-determining programs that considered different combinations of cell size, [SPCH], signaling neighbours, and/or their interaction terms (Fig. 7a, right). The top model was one where degradation rates scaled with neighbours, and where size, [SPCH], and neighbours were all considered in the fate-determining process ("best model overall"; Fig. 7b, Supplementary Fig 7a, b). The fact that a fate-determining program with size, [SPCH], and neighbours outperformed one with [SPCH] alone implies that there are SPCH-independent pathways through which size and neighbours operate (Fig. 8).

## Discussion

In this study, we leverage quantitative approaches to define the properties and behaviours of the enigmatic stem-cell-like SLGCs in the *Arabidopsis* leaf epidermis. We show that the division potential of an SLGC is a product of its neighbourhood and nuclear size. Larger cells divide less often because they have more signaling neighbours and larger nuclei, resulting in lower [SPCH]. Through computational simulations, we also identify SPCH-independent pathways that may link cell size and signaling to behaviour.

In a previous study, we showed that cell size regulates meristemoid behaviour: larger meristemoids divide more often than smaller ones[15]. Our findings here extend that work in two important ways. First, we make the surprising observation that while cell size is also associated with SLGC behaviour, the direction of this relationship is inverted: larger SLGCs divide *less* often than smaller ones. Second, by linking cell size to the dynamics of SPCH, we can propose a molecular explanation for why large SLGCs divide less often. More broadly, our work expands on existing studies of cell size and behaviour[15,36,37], which have focused primarily on cell-autonomous factors, to consider the neighbourhood of a cell. We demonstrate that cell size can affect non-cell-autonomous factors, such as the strength of cell-cell signaling.

There is a growing recognition of the importance of cell geometries for signaling[38,39]. For example, Pentinmikko et al.[40] used in vitro organoids and culture scaffolds to show that the area of neighbour contacts in small intestinal stem cells affects the strength of signals they receive. In these stem cells, apical constriction increases the lateral surface-to-volume ratio, which enhances their ability to receive niche signals from neighbouring Paneth cells. When the lateral surface-to-volume ratio was reduced, the stem cells initiated fewer organoids, suggesting their regenerative capacity was disrupted[40]. Similarly, we report here that the size of an SLGC can affect the magnitude of the signals it receives, as measured by the rate of SPCH degradation.

In the leaf epidermis, stomata and their precursors (meristemoids and GMCs) are thought to secrete mobile peptides to prevent their neighbours from dividing asymmetrically and producing stomata. This ensures that stomata are spaced apart, which optimizes stomatal opening and environmental responsiveness[30]. Unlike meristemoids, stomata and GMCs are committed cells that cannot respond to an adjacent stomatal precursor by dividing asymmetrically. Thus, one would expect the strongest inhibitory signals to come from these cells. Surprisingly, we find that meristemoids have the strongest effect on SLGC division potential among the three cell types. Signaling among progenitors may be critical to regulate cell numbers during the proliferative phase of leaf development. It is also possible that we have underestimated the impact of stomata because very few SLGCs have stomatal neighbours at this early developmental stage.

In this work we focused primarily on the influences coming from the immediate neighbourhood of an SLGC. It is also important to consider whether SLGC behaviour is influenced by factors operating at larger spatial scales. As a proxy for tissue-wide effects, we included each SLGC's position in the leaf—the X- and Y-coordinates—in our original

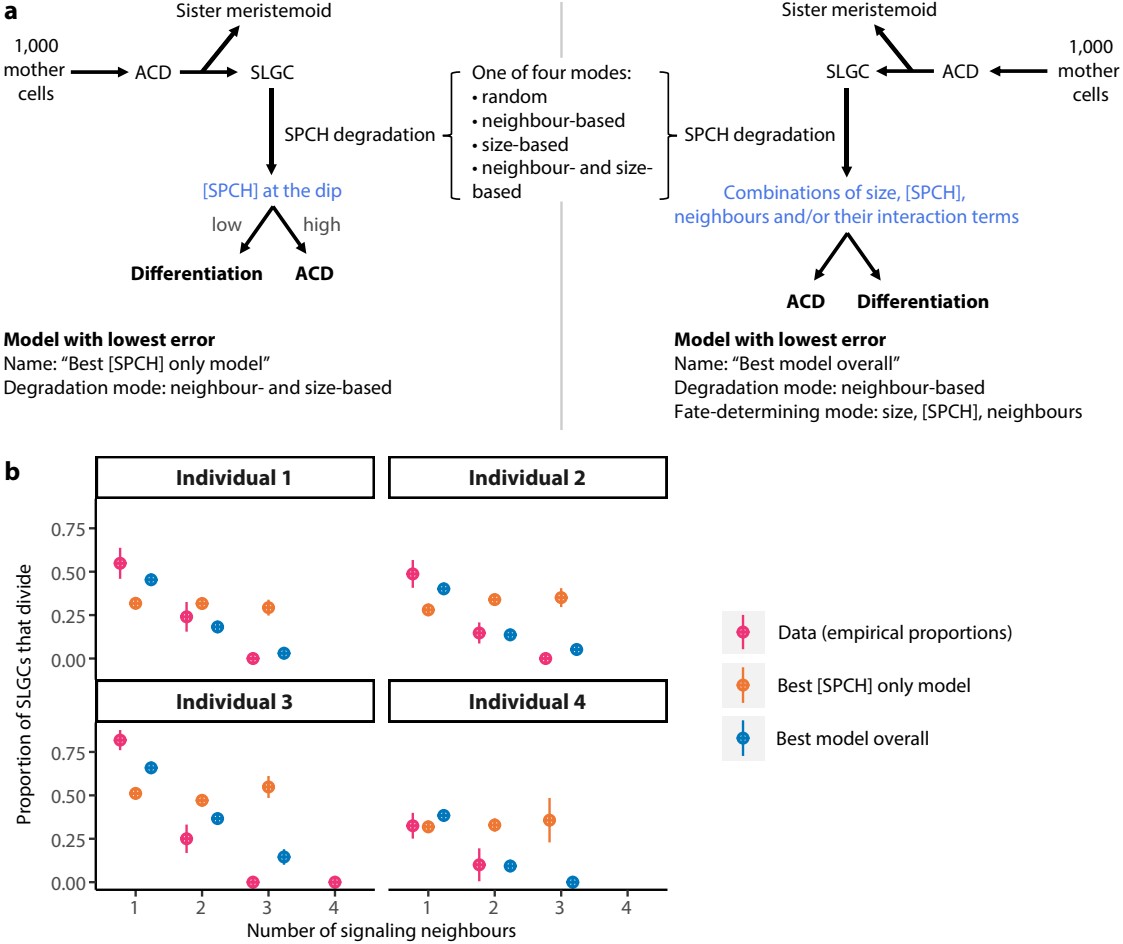

**Fig. 7 | The best performing model for fate-determination considers size, [SPCH], and signaling neighbours, and outperforms one with [SPCH] alone.**
**a** Simplified diagram of the lineage decision tree model. The version on the left is one where [SPCH] at the dip is the sole determinant of SLGC behaviour. We tested different degradation modes: the degradation mode that yielded the lowest error was one where degradation rates scaled with both signaling neighbours and size ("Best [SPCH] only model"). To determine whether other features were required for fate determination, we specified additional fate-determining modes that considered different combinations of cell size, [SPCH], signaling neighbours, and/or their interaction terms (right). The best performing model was one where degradation rates scaled with neighbours, and where size, [SPCH], and neighbours were all considered in the fate-determining process ("Best model overall"). **b** The proportion of SLGCs that divided given the number of signaling neighbours in the experimental data (pink), Best [SPCH] only model (orange), and Best model overall (blue). Circles and lines are means and standard deviations. Data: $N = 4$ individual plants; 50, 62, 75, 75 cells per individual. Models: $N = 4$ individual plants; 1000 cells per individual. Source data are provided in Source Data file 1.

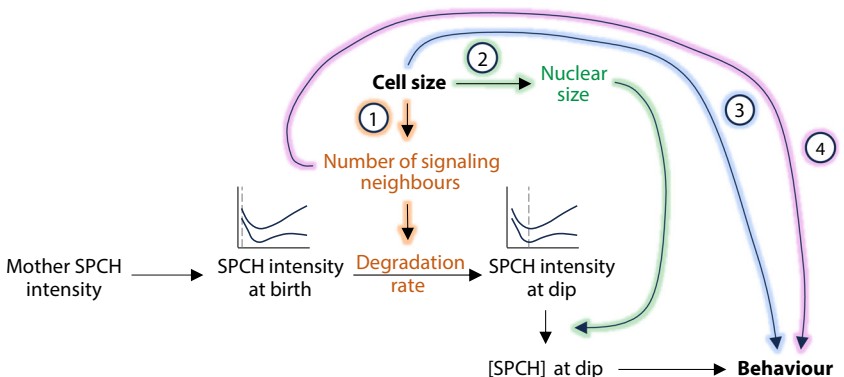

**Fig. 8 | Size is linked to SLGC behavior through SPCH degradation, SPCH concentration and a SPCH-independent pathway.** Model for SLGC division propensity. Cell size is linked to SLGC behaviour through (1) SPCH degradation, gold; (2) SPCH concentration, green; and (3) a SPCH-independent pathway, blue. (4) Based on our simulations (Fig. 7, Supplementary Figs 5 and 7; Supplementary Tables 2-4), signaling neighbours may affect behaviour in a SPCH- and size-independent manner, purple.

decision tree analysis (Fig. 1d, Supplementary Fig. 1a). Neither coordinate was predictive of SLGC behaviour, which indicates that the non-cell-autonomous factors that govern SLGC behaviour are primarily local.

We showed that the SPCH dynamics in dividing and differentiating cells bifurcate in early G1 (~200 min after birth; Fig. 2), which raises the question of whether the decision to divide is made at this point. The dynamics of CDT1A, a replication licensing factor, appear to support this notion (Supplementary Fig. 2a). In dividing cells, CDT1A begins to accumulate in early G1, approximately 300 minutes after birth (Supplementary Fig. 2a). Since CDT1A accumulation is a hallmark of cell cycle progression[21], these dynamics suggest that the decision to divide is made in early G1, and no later than ~300 minutes after birth.

An elegant G1 cell size sensing mechanism has been described for the self-renewing divisions in the *Arabidopsis* shoot apical meristem[41]. Here, cells of different sizes inherit equal amounts of KRP4, a cell cycle repressor, via KRP4's stoichiometric association with chromatin. Smaller cells must grow more than larger cells to dilute KRP4 sufficiently to allow passage through G1 phase, and thus this mechanism can maintain cell-size homeostasis in the population[41]. We proposed that SLGCs use a dilution-based size sensing mechanism, and a natural question is whether KRP4 or its homologues are involved in the SPCH-independent activity (Fig. 8, pathway 3). We think it is unlikely because in SLGCs, smaller cells are more likely to divide, and thus a cell cycle activator, rather than an inhibitor, would need to be the inherited and diluted factor.

There is broad precedent for cell fate decisions in early G1. For example, human embryonic stem cells can only differentiate into endoderm if they receive TGF-β-Smad2/3 signals in early G1, when cyclin D levels are low[42]. Once cyclin D levels rise in late G1, the Smad2/3 proteins are phosphorylated, which prevents them from entering the nucleus and activating endoderm genes[42]. Likewise, the decision to undergo a formative or proliferative division in *Arabidopsis* roots is thought to occur in early G1[14]. Cells expressing low, transient levels of the transcription factors SCARECROW and SHORTROOT in early G1 are likely to divide formatively, rather than proliferatively[14].

Interestingly, in cells that might be the most closely related to SLGCs, the giant cells of the sepal epidermis, cell fate decisions are linked to a concentration threshold of the HD-ZIP transcription factor AtML1 in G2[43]. Two elements of the Meyer study provide useful context and contrasts to our work. Giant cells are highly endoreplicated, and differentiation of SLGCs into pavement cells is also accompanied by endoreplication. Models of AtML1 function suggest that its G2 expression enables regulation of genes that promote endoreplication over mitotic cycles in giant cells[43]. Our finding that [SPCH] in early G1 is predictive of SLGC fate would suggest that endoreplication is a secondary consequence, rather than a cause of differentiation. Second, Meyer's work emphasizes the cell-autonomous nature of the giant cell fate choice, which fits into their mechanical role in sepal shape. SLGCs, on the other hand, are sensitive to non-cell-autonomous factors, which enables them to carry out the stem-cell-like function of modulating leaf cell numbers and types in response to external inputs.

Considered together, our quantitative imaging, statistical, simulation, and experimental approaches identified some of the key players in the SLGC decision, including cell size, SPCH activity, and cell-cell signaling. Our work shows how initial differences in SPCH levels are canalized by nuclear size and signaling to inform flexible cell fate decisions. It also highlights the existence of SPCH-independent pathways that link cell size and signaling to behaviour, which will be an important avenue of study moving forward.

## Methods

### Plant material and growth conditions
All *Arabidopsis* lines were in the Col-0 background ("wild-type"). Seeds were surface-sterilized by ethanol or chlorine gas[44] and stratified for two days. Following stratification, they were grown vertically on half-

strength Murashige and Skoog (MS) media with 0.8% or 1% agar for five days under long-day conditions (16 h light: 8 h dark at 22 °C) and moderate intensity, full-spectrum light (110 μE).

Newly generated and previously reported lines are described in Supplementary Table 5. All transgenes used have been reported previously. Transgenic lines were generated by floral dip[44] and transgenic seedlings were selected on half-strength MS with the appropriate antibiotic (50 mg/L kanamycin or hygromycin). Seeds of transgenic lines are available upon request from corresponding author, dbergmann@stanford.edu.

### Image acquisition and image analysis
All confocal imaging experiments were performed on a Leica SP5 or Stellaris 8 confocal microscope with HyD detectors, a 40x NA1.1 water objective, image size of 1024 × 1024 pixels, and a digital zoom of 0.75-1x (unless otherwise specified). Only the abaxial surfaces of cotyledons were imaged. Raw Z-stacks were projected with Sum Slices in Fiji[45].

### Wild-type time-lapses
To explore the relationship between birth size and SLGC behaviour, we captured time-lapses of 3-dpg, wild-type seedlings bearing a plasma membrane marker (*pATML1::mCherry-RCI2A* or *pATML1::YFP-RCI2A*) and a polarity marker (*pBRXL2::BRXL2-YFP*), a nuclear marker (*pATML1::H2B-mTFP*), or a cell cycle marker (PlaCCI)[46]. Seedlings were mounted in a custom imaging chamber with half-strength MS solution[33] and imaged at 40 or 45 min intervals for ~16 h. Cell size and shape were extracted from ilastik segmentations of the plasma membrane channel (Fig. 1c)[15,16]. Features "at birth" were measured from the first frame in which the newly formed cell plate was visible. After imaging, seedlings were returned to half-strength MS agar plates, where they were grown under long-day conditions (16 h light:8 h dark at 22 °C) and moderate intensity light (110 μE). Two days later, they were re-imaged to capture the developmental outcomes of SLGCs and their neighbours. Repeated imaging of seedlings must be done carefully to avoid damage and growth arrest. We only kept a time-lapse for analysis when meristemoids, SLGCs and GMCs underwent an equivalent number of cell divisions between 3-5 dpg as those same classes of cells from sibling plants that had not been subjected to long term time-lapse imaging at 3-dpg (e.g. they had only a single imaging session at 3-dpg), and when the overall organ sizes of imaged and sibling plants were similar. These stringent quality controls meant that we only used 25% of all time-lapse experiments generated.

### CDT1A intensities
To quantify CDT1A intensities, we enclosed each SLGC nucleus in the PlaCCI time-lapse described above in a circular ROI (area: 45.28 μm²) and measured the background-subtracted raw integrated density of CFP within each ROI.

### SPCH reporter time-lapses
To quantify SPCH levels in SLGCs, we acquired time-lapses of 3-dpg seedlings expressing a plasma membrane marker (*pATML1::mCherry-RCI2A*) and a SPCH translational reporter (*pSPCH::SPCH-YFP* rescuing *spch-3*). Individuals were imaged as described above, except for one individual (#4), which was mounted on a slide with vacuum grease and imaged for 8 h at 60 min intervals. To quantify SPCH intensities, we segmented the plasma membrane channel using ilastik[16] and measured the background-subtracted raw integrated density of YFP within the cell boundaries of each SLGC.

Because our SPCH reporter line lacked a genetically encoded nuclear marker, we could not measure nuclear concentrations directly from our data. Instead, we stained the nuclei of 3-dpg seedlings with Hoechst[15], segmented both the nuclear and genetically encoded plasma membrane signals using ilastik[16], and fitted a linear regression model to our ln-transformed cell and nuclear area measurements

($\alpha = 1.82 \pm 0.14$, $\beta = 0.19 \pm 0.032$, $t = 5.83$, $P = 3.42$e-08; Supplementary Fig. 2b). Our model was not heteroscedastic (Breusch-Pagan test: $\chi^2 = 0.034$, $P = 0.85$), so we assumed that the distribution of residuals at any point along the fitted line could be modeled as a normal distribution $N(6.45$e-18, $0.027)$. For each cell in our SPCH dataset, we estimated nuclear area from this equation: $ln$(nuclear area) = 0.19 * $ln$(cell area) + 1.82 + $\epsilon$, where $\epsilon \sim N(6.45$e-18, $0.027)$. We divided SPCH intensity by nuclear area to obtain nuclear concentrations.

## Analysis of the contribution of bleaching to SPCH behaviours

To determine whether bleaching alone could account for the observed decline in SPCH intensities, we estimated bleaching rates per individual seedling. Using the Fiji plugin TrackMate[47], we quantified the background-subtracted raw integrated density of YFP in each nucleus of each frame of the time-lapse. We regressed intensity on time (in hours) and divided the slope by the intercept to estimate a bleaching rate in % per hour. To compute the overall rates of decline in SPCH intensities, we fit an exponential decay function ($N(t) = N_0 * e^{\lambda t}$) to the SPCH intensities in each cell from 0 to ~200 minutes after birth and estimated the decay constant $\lambda$. We computed the overall rate of decline (% SPCH lost per hour) as $100\% * (1 - e^{\lambda})$. Bleaching rates were low, ranging from 1.5-2.2% per hour, compared to mean rates of decline of 23-42% per hour (Supplementary Table 1).

## SPCH++ time-lapses

To test whether large cells divide when supplied with enough SPCH, we captured time-lapses of *spch-3* seedlings expressing a plasma membrane marker (*pATML1::mCherry-RCI2A*) and a transgene *pSPCH::SPCH-YFP* that overproduces SPCH in its native domain. SPCH overproduction was verified through phenotypic analysis (an increase in asymmetric cell divisions; Fig. 3a) and fluorescence quantification (Supplementary Fig. 3d). We imaged three individuals: one individual was imaged in the custom time-lapse chamber[33] and the remaining two were imaged on slides with vacuum grease at 0, 3, and 6 hours, before being returned to half-strength MS plates[15]. Two days later, they were re-imaged to capture cell behaviours. SLGCs grow very slowly (mean ± standard deviation: 1.46 ± 0.88% per hour), so this modified protocol only increased the error in birth size measurements due to cell growth from ~1 to ~3%, while enabling a larger number of individuals to be imaged simultaneously.

## SPCH2-4A time course

Three *spch-3* seedlings expressing a plasma membrane marker (*pATML1::mCherry-RCI2A*) and *pSPCH::SPCH2-4A-YFP*[33] were imaged once at 3-dpg and again at 5-dpg, using 25X and 40X objectives. Cell sizes were extracted from ilastik segmentations of the plasma membrane channel at 3-dpg[16].

## CYCD7-YFP time course

A seedling expressing a plasma membrane marker (*pATML1::mCherry-RCI2A*) and *pATML1::CYCD7-YFP* was imaged once at 3-dpg and again at 5-dpg. Cell sizes were extracted from ilastik segmentations of the plasma membrane channel at 3-dpg[16]. In addition to imaging the entire cotyledon at a digital zoom of 0.75x (Supplementary Fig. 3a), we also imaged one region at 4x (Supplementary Fig. 3b).

## Measuring fraction of the cell perimeter in contact with a signaling neighbour

We calculated the fraction of the cell perimeter in contact with a given signaling neighbour as

$$0.5*\left(P_{SLGC} + P_{neighbour} - P_{union}\right)/P_{SLGC}$$

where $P_{SLGC}$ is the perimeter of the SLGC, $P_{neighbour}$ is the perimeter of the signaling neighbour, and $P_{union}$ is the perimeter of the union of the SLGC and the signaling neighbour. The total fraction of the cell perimeter in contact with a signaling neighbour is the sum of these fractions.

## Statistical analysis

**Classification and regression tree.** To identify correlates of SLGC behaviour, we measured 15 features of SLGCs at birth (listed in Supplementary Fig. 1a) in a seedling expressing a plasma membrane marker (*pATML1::mCherry-RCI2A*) and a cell cycle marker (PlaCCI)[46]. We fed our measurements into a classification and regression tree (CART), which we implemented in Python using the *scikit-learn* library (sklearn.tree module)[48]. The CART method builds a decision tree by recursively partitioning cells along predictor axes into subsets that divide or differentiate. We first split our data into training and test sets (70:30) by randomly sampling without replacement. Next, we fit the CART algorithm to the training set, using Gini impurity as a measure of split quality. The Gini impurity was calculated as

$$\sum_{i=1}^{C} p(i)*(1 - p(i))$$

where $C$ is the total number of classes (two in our case: division & differentiation) and $p(i)$ is the probability of selecting a datapoint in class $i$. The lower the Gini impurity, the more accurate the split. We then followed the "Post pruning decision trees with cost complexity pruning" protocol from *scikit-learn* to prune the decision tree to prevent overfitting[48]. We chose a cost complexity parameter value (ccp_alpha) that maximized testing accuracy.

We also specified a random forest classifier (sklearn.ensemble module), which controls overfitting by fitting 1000 trees to various subsamples of the data and computing an average prediction accuracy. Feature importance scores were calculated as the normalized, total reduction of Gini impurity contributed by a given feature[48].

**Analyses in R.** Mixed-effects models were specified using the *nlme* package (*v3.1-162*)[49] with predictors of interest as fixed effects and individual as a random effect. All other comparisons were made using two-sample tests for equality of proportions with continuity correction (*stats* package v.4.3.1)[50], unpaired two-sided Wilcoxon rank sum tests with continuity correction (*stats* package v.4.3.1)[50], or chi-squared tests for trend in proportions (*rstatix* package v0.7.2)[51]. Exponential, linear, and logistic models were fit with the *stats* package (*v4.3.1*)[50], with individual as a blocking variable where appropriate. Additional information about the statistical tests performed for each figure panel is reported in the Source Data file.

**Simulations.** The lineage decision tree model was implemented in MATLAB 2021a and expands on the model reported in Gong, Dale, Fung, Amador et al.[15]. It is a stochastic, asynchronous rule-based model where SLGCs undergo birth, growth, SPCH degradation, and differentiation or division (Fig. 7a, Supplementary Fig. 5). Cell sizes were rounded to the nearest integer $\mu m^2$. Nuclear sizes were estimated as described above: $ln$(nuclear area) = 0.19 * $ln$(cell area) + 1.82 + $\epsilon$, where $\epsilon \sim N(6.45$e-18, $0.027)$.

The starting sizes, SPCH intensities, and numbers of signaling neighbours of 1000 mother cells were randomly drawn from gamma, normal, and Poisson distributions respectively, fitted to empirical distributions from each of four individual plants (Supplementary Fig. 5). The cells then divided with an asymmetry parameter (1 − SLGC size/mother size) drawn from beta distributions fitted to empirical distributions from each of four individual plants (Supplementary Fig. 5). A uniformly distributed noise factor of ±0.5 was applied to the asymmetry parameter. Based on Supplementary Fig. 6, each SLGC was assumed to inherit two-thirds of its mother's SPCH intensity and to have one signaling neighbour more than its mother (i.e. the newly

generated sister meristemoid). SPCH was degraded according to one of four degradation modes:

- Random (neighbour- and size-independent): decay constants (λ) were randomly drawn from an exponential distribution fitted to the empirical decay constant distribution of all cells in an individual
- Neighbour-based degradation: decay constants were randomly drawn from one of two exponential distributions, fitted to the empirical decay constant distributions of cells with 1 and 2+ neighbours, respectively (see Supplementary Fig. 5)
- Size-based degradation: per-square-micron decay constants (λ/ cell size) were randomly drawn from an exponential distribution fitted to the empirical per-square-micron decay constant distribution
- Neighbour- and size-based degradation: per-square-micron decay constants were randomly drawn from one of two exponential distributions, fitted to the empirical per-square-micron decay constant distributions of cells with 1 and 2+ neighbours, respectively

The probability of division was determined based on a cell's size, [SPCH] at the dip, number of signaling neighbours, and/or their interaction terms, using multiple logistic parameters estimated via the *stats* package in R (*v4.3.1*; Supplementary Table 3)[50].

Model selection was performed by simulating across modes of SPCH degradation (random, neighbour-based, size-based, or neighbour- and size-based) and modes of fate determination (including linear and interaction terms of cell size, [SPCH], and the number of signaling neighbours) in a factorial manner. Simulations were run for one generation. Model selection occurred in three steps. First, the sizes and [SPCH] of dividing and differentiating cells were compared to those of the data using "two one-sided tests" (TOST) equivalence testing. The null hypothesis in TOST equivalence testing is that there is a difference in populations greater than the effect size of interest. Due to our experimental sample size, we chose an effect size of one standard deviation[52]. We used Welch's *t*-tests for unequal sample sizes with the Satterthwaite correction. A significance threshold of 0.05 was used to determine equivalence.

Second, two-sample *t*-tests were run to check if there were significant differences in the sizes and [SPCH] of cells that divided vs. differentiated. A threshold of 0.05 was used to determine significance. Finally, the sum of squared errors (SSE) was calculated to assess the fit of each model to the proportion of cells that divided given the number of signaling neighbors. The corrected Akaike information criterion (AICc) was calculated for the total SSE across individuals using the following formula:

$$AICc = N \times \log\left(\frac{\sum SSE}{N}\right) + 2(k+1) + \left(2k^2 + 2k\right)/(N - k - 1),$$

where $N$ is the total number of cells in the data (262) and $k$ is the number of parameters in the fate-determining logistic models, excluding the intercept.

## Reporting summary
Further information on research design is available in the Nature Portfolio Reporting Summary linked to this article.

## Data availability
All data needed to evaluate the conclusions in the paper are present in the paper and/or in a Source Data File. Source data are provided with this paper.

## Code availability
Code for simulations is available at Figshare: https://doi.org/10.6084/m9.figshare.27931686.v1.

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

## Acknowledgements

We thank Madhav Mani (Northwestern), lab members and the Stanford cell size control community for engaged discussions. We thank Adrienne Roeder (Cornell) and former Bergmann lab members Anne Vatén, Kelli Davies and Juliana Matos for reagents. H.F.F. was supported by a graduate fellowship award from Knight-Hennessy Scholars at Stanford University. G.O.A. was supported by funds from the National Institutes of Health (T32 5T32GM007790), the National Science Foundation (DGE-1656518), and a Stanford Graduate Fellowship. R.D. is supported by an NSF postdoctoral fellowship (IOS-2109790). M.V. and J.M.E. were supported by funds from the National Institutes of Health (T32GM007276).

## Author contributions

H.F.F. led conceptualization, data acquisition, analysis and interpretation and wrote the manuscript. G.O.A. contributed to data acquisition, analysis and interpretation. R.D. created and analyzed computational models for SLGC behavior. Y.G., M.V., J.M.E. and A.M. created reagents and contributed to data acquisition and analysis. D.C.B. supervised the work.

## Competing interests

The authors declare no competing interests.
