## [Transparent Peer Review file · Nature Communications]

Multi-scale dynamics influence the division potential of stomatal lineage ground cells in Arabidopsis

Corresponding Author: Professor Dominique Bergmann

Version 0:

Reviewer comments:

Reviewer #1

(Remarks to the Author)

The paper by Fung et al addresses a fundamental question in cell biology- what controls the decision of a cell to divide? As an experimental system they use stomata patterning in the plant epidermis which provides a strong foundation of knowledge and tools to investigate the problem. The topic and data will be of broad interest to the readership of Nature Communications.

The authors make a compelling case that the nuclear concentration of a key transcription factor (SPCH) influences the decision of continued cell division (or not), with new information/ideas on how the number of neighboring cells influences the process (via a signalling process that is yet to be fully elucidated), plus the finding that not all neighboring cells seem to be equal as a signalling source (with meristemoids having a stronger influence than other neighboring cells). Finally the authors put the data into a modelling context which indicate that the SPCH system alone is insufficient to explain all the cell behaviours observed. These are important observations, based on a strong experimental approach, that advance the field.

In addition to these positive comments, I have some questions that the authors might address:

- 1) As indicated in the discussion, there is interesting work in the research area relating cell volume/surface area and local signalling for differentiation. The work here only measures cell surface area, not volume. There is, therefore, the assumption that cell depth is more or less constant so that cell area can be used as a proxy for volume (cell size). In the context of the cotyledon epidermis this is a reasonable assumption (I would not expect it to vary massively), but if the authors have any data to confirm this, it would be good to include.
- 2) The experimental procedure involves moving samples from a growth chamber to the confocal, then (after viewing) return to a chamber, followed (after another time interval) return to the confocal for imaging. This is technically tricky with the risk of the manipulation itself influencing observations made. The authors are commendably upfront and indicate (line 357) that only c. 25% of experiments were used. Can they be more explicit on how they decided which data sets to exclude? As indicated above, the "success" rate sounds very plausible, but it would be very useful to have more detail on the criteria used to include/exclude data sets. Much of the interpretation of the data requires a probabilistic analysis, so there is always the risk that inadvertent data selection might skew outcomes.
- 3) In most of the paper the authors report relationships in SPCH intensity vs in SLGC area, (e.g., Fig 2G,H) but in Fig S3D (looking at lines where SPCH is overexpressed) they show the untransformed data. Is there a reason for this?
- 4) Linked to point (3), there is, not unsurprisingly, a very wide spread of data for, e.g., SPCH intensity. Does the fact that this can be interpreted as a linear model after transformation by ln tell us something about the nature of the interaction?
- 5) The authors looked at 4-6 biological representatives per genotype, yet on the graphs I think that individual data points represent individual cells (i.e., collated data from all biological representatives and analysis was done on the collated data set). Have I understood this correctly? Or did they perform the analysis on collated data points (cells) from each biological replicate separately, then examine how any patterns observed were conserved across the 4-6 replicate data sets? This was unclear to me and could have repercussions on data interpretation if the former approach was taken (how independent are n individual cell sizes in 1 biological replicate)?

6) One of the interesting observations made is that the size of meristemoids show a positive correlation with propensity for division (previous papers) whereas here the size of the SLGC shows an inverse correlation with propensity for division. Could this relate to a switch in the type of cell growth? As plant cells differentiate and grow larger, this is generally driven by increase in vacuole size, whereas meristemoid growth seems to be more driven by cytoplasmic increase. Are there any extant data comparing the SLGCs at this early stage in terms of vacuole/cytoplasm volume ratios? This isn't required for publication, more a line of thought that might be included (if it holds water).

7) In the summary cartoon (Fig 6A) there is no feedback loop from number of neighboring signalling cells and cell size, yet when I was reading the text my impression was that this going to be an important element of the system (i.e., SLGC size influences the number of neighbours and number (and type) of neighbours influences SLGC size. Perhaps I have misunderstood, but it would seem reasonable for such a feedback system to be in place.

8) In the references I couldn't see a reference to the work of D'Ario et al (2021) DOI: 10.1126/science.abb4348 which describes how the nuclear level of a KRP protein influences cell size control in plant stem cell niches. Is there a potential crossover here of the two systems (SPCH and KRP)? Might be worth including in the discussion. Could this be the "SPCH-independent" required by their model? Or could they both be acting in the same pathway?

Reviewer #2

(Remarks to the Author)

Review of the manuscript: "Multi-scale dynamics influence the division potential of stomatal lineage ground cells in Arabidopsis."

The authors investigate the key question of how the critical decision—whether lineage precursors differentiate or continue dividing—is regulated, using stomatal lineage ground cell (SLGC) development in Arabidopsis. They employ various methods, such as long-term imaging and statistical modeling, to identify features that influence SLGC behavior at different spatiotemporal scales (including time, subcellular, and intercellular levels). Their findings emphasize the size of cells and SPCH concentration and cell-cell signaling, particularly from neighboring stem cells like meristemoids, in determining SLGC behavior. Interestingly, this decision-making process in SLGCs contrasts with that of meristemoids, as the authors' group previously discovered.

Given the challenges of studying the factors and transcriptomes involved in SLGC characteristics, as well as the limited available information, the discoveries and approaches used in this study—particularly the extensive imaging analyses and modeling—offer valuable insights to help bridge the knowledge gap in SLGC fate decisions during epidermal cell development.

I have a few comments on the manuscript.

It would be better to emphasize the importance of both cell size and SPCH concentration equally, as well as the existence of SPCH-independent pathways in SLGC fate determination. While the findings from the imaging analyses were sufficiently highlighted, the modeling results were underrepresented in the main figures. To address whether SPCH alone is sufficient to explain SLGC behavior, the authors used modeling. They discovered the existence of SPCH-independent pathways that link cell size and signaling to SLGC behavior, which is one of the key highlights of their work. However, this is not mentioned in the abstract, and the detailed data are missing from the main figures.

Figure 5B: The Y-axis title should be consistent across both graphs, and an explanation of the p-value is missing from the figure legend.

Figure 6: It would make more sense for "(A) Model for SLGC division propensity" to be placed after (B) and (C).

Lastly, the reference "Zeileis, A., Hothorn, T., 2002. Diagnostic Checking in Regression Relationships" appears to be missing from the text.

Reviewer #3

(Remarks to the Author)

The manuscript studies the fate decisions of stomatal lineage ground cells (SLGC) in Arabidopsis, specifically whether they differentiate or divide asymmetrically. The authors report that large cells tend to differentiate and propose a link between division behaviour and the transcription factor SPEECHLESS (SPCH) both at the neighbourhood scale (strength of cell-cell signalling) and at the subcellular scale (size of the nucleus and thus SPCH nuclear concentration). Using experimental and computational approaches, they propose specific decision pathways leading to differentiation or division based on cell size, number of signalling neighbours and SPCH levels.

The manuscript is well written with a good flow and a nicely presented story, with some room for improvement.

1. The main point to mention is the need for more explanations and details on the statistical and computational methods. For

example, terms such as Gini impurity and cost complexity pruning may be unfamiliar to people outside the field (even for applied mathematicians working on modelling and model selection).

2. Some additional background on Arabidopsis development necessary to follow the manuscript would be useful to understand, for example, why 3 days post germination was chosen.

3. Line 227: it is not clear what is meant by randomly drawn asymmetry? Is it regarding the cell sizes of the daughter cells?

4. The lineage decision tree diagram is helpful, but a flowchart with more detail and actually showing where each decision is made, how the degradation rate is modulated, and at what point the parameters are adjusted would be even more helpful.

5. Also, it is not clear at first glance whether the cells are actually simulated independently of each other, with no relationship between the 1000 mother cells and all properties drawn from specific distributions, or whether real space is also included? Again, better explanations with clearer details about methods and simulations are needed, and a flowchart would be very useful.

6. Regarding the AIC: depending on the sample size of the data, the corrected AIC (AICc) might be more accurate. In addition, it seems that the AIC values are quite similar and it is not easy to determine with certainty whether the top model is the best. In addition, Akaike weights could be calculated to see how likely each model is.

7. Are the data and code available?

8. Supplementary Figure 4C is confusing. The probabilities of division are either close to 0 or close to 1 and a logistic regression is fitted to this. It is also very difficult to see the colours of the dots. What are we trying to show here?

9. Figure 2A-B: how is it established that the cell in A divides and that in B differentiates? Could you show this with images from a later time step?

Minor comments:

10. The term "dip" is not clearly defined in the main text.

11. On line 185: mention that total number of neighbours means signalling and non-signalling.

Version 1:

Reviewer comments:

Reviewer #1

(Remarks to the Author)

The authors have addressed the points raised in my review.

(Remarks on code availability)

Reviewer #3

(Remarks to the Author)

The authors have addressed all my concerns. In my opinion the manuscript can be now accepted for publication.

(Remarks on code availability)

RESPONSE TO REVIEWER COMMENTS

Reviewer #1 (Remarks to the Author):

The paper by Fung et al addresses a fundamental question in cell biology- what controls the decision of a cell to divide? As an experimental system they use stomata patterning in the plant epidermis which provides a strong foundation of knowledge and tools to investigate the problem. The topic and data will be of broad interest to the readership of Nature Communications.

The authors make a compelling case that the nuclear concentration of a key transcription factor (SPCH) influences the decision of continued cell division (or not), with new information/ideas on how the number of neighboring cells influences the process (via a signalling process that is yet to be fully elucidated), plus the finding that not all neighboring cells seem to be equal as a signalling source (with meristemoids having a stronger influence than other neighboring cells). Finally the authors put the data into a modelling context which indicate that the SPCH system alone is insufficient to explain all the cell behaviours observed. These are important observations, based on a strong experimental approach, that advance the field.

In addition to these positive comments, I have some questions that the authors might address:

1) As indicated in the discussion, there is interesting work in the research area relating cell volume/surface area and local signalling for differentiation. The work here only measures cell surface area, not volume. There is, therefore, the assumption that cell depth is more or less constant so that cell area can be used as a proxy for volume (cell size). In the context of the cotyledon epidermis this is a reasonable assumption (I would not expect it to vary massively), but if the authors have any data to confirm this, it would be good to include.

This is an important point. The epidermal cell “depth” is fairly uniform in the cell types and growth stages we measured, something we and others have noted in passing in other papers (Wuyts et al., 2010 <https://doi.org/10.1186/1746-4811-6-17>, Dow et al., 2017, <https://doi.org/10.1111/nph.14746>). And we have now explicitly mentioned this in the text (line 86). An example of the “depth” of the epidermal cells at 3-dpg is shown below. The apical (outer) surface is at the top.

This image is created by taking source data from one of the WT timelapses used in the current paper. We always set up the timelapse Z-stacks to capture the region from the surface into the mesophyll, thus we have the entirety of the epidermal cells. The image here was created via Fiji commands Image > Stacks > Reslice. Meristemoids, SLGCs, guard cells and young pavement cells are all included in the image shown here.

2) The experimental procedure involves moving samples from a growth chamber to the confocal, then (after viewing) return to a chamber, followed (after another time interval) return to the confocal for imaging. This is technically tricky with the risk of the manipulation itself influencing observations made. The authors are commendably upfront and indicate (line 357) that only c. 25% of experiments were used. Can they be more explicit on how they decided which data sets to exclude? As indicated above, the “success” rate sounds very plausible, but it would be very useful to have more detail on the criteria used to include/exclude data sets. Much of the interpretation of the data requires a probabilistic analysis, so there is always the risk that inadvertent data selection might skew outcomes.

Our main criteria were growth at the whole organ level and growth and division at the cellular level. For the experiment, the leaves are being imaged actively for 6-16 hours, and then returned to nutrient plates for two days. We discarded a timelapse imaging experiment when (1) meristemoids, SLGCs and guard mother cells (GMCs) failed to undergo a roughly equivalent number of cell divisions as those from plants that had only undergone a short-term initial round of imaging and/or (2) there was minimal leaf or cell growth in the 2 days after the initial imaging. We have added these details about what criteria we considered when keeping or discarding a timelapse experiment to the methods (starting line 357).

3) In most of the paper the authors report relationships \ln SPCH intensity vs \ln SLGC area, (e.g., Fig 2G,H) but in Fig S3D (looking at lines where SPCH is overexpressed) they show the untransformed data. Is there a reason for this?

Thank you for catching that—the data in Fig. S3D are now \ln -transformed.

4) Linked to point (3), there is, not unsurprisingly, a very wide spread of data for, e.g., SPCH intensity. Does the fact that this can be interpreted as a linear model after transformation by \ln tell us something about the nature of the interaction?

Cell size, SPCH intensity, and SPCH concentration are right-skewed (see panels A-C below). The natural log transformation makes the data more normally distributed, which is more appropriate for a linear regression: it improves the variance, reducing heteroskedasticity. The observation that cell size and SPCH concentration are only linearly related after transformation suggests they show a power scaling relationship due to the effect of dilution at large cell sizes.

5) The authors looked at 4-6 biological representatives per genotype, yet on the graphs I think that individual data points represent individual cells (i.e., collated data from all biological representatives and analysis was done on the collated data set). Have I understood this correctly? Or did they perform the analysis on collated data points (cells) from each biological replicate separately, then examine how any patterns observed were conserved across the 4-6 replicate data sets? This was unclear to me and could have repercussions on data interpretation if the former approach was taken (how independent are n individual cell sizes in 1 biological replicate)?

The P -values were calculated using mixed-effects models, which can be applied to hierarchical experimental designs such as ours, with one level being the individual seedlings/replicates and the second being the individual cells. By treating “replicate” as a random effect, the model accounts for the non-independence of the cell-level measurements.

6) One of the interesting observations made is that the size of meristemoids show a positive correlation with propensity for division (previous papers) whereas here the size of the SGLC shows an inverse correlation with propensity for division. Could this relate to a switch in the type of cell growth? As plant cells differentiate and grow larger, this is generally driven by increase in vacuole size, whereas meristemoid growth seems to be more driven by cytoplasmic increase. Are there any extant data comparing the SGLCs at this early stage in terms of vacuole/cytoplasm volume ratios? This isn't required for publication, more a line of thought that might be included (if it holds water).

This is an interesting idea to consider in the future. In our previous work (doi.org/10.1126/sciadv.adf3497) we had been concerned about the impact of the vacuole when considering cell size relationships. We were fortunate that we could use careful electron tomography measurements from Liwen Jiang's lab to see that vacuoles are of fairly uniform size and number in meristemoids (doi.org/10.1093/plphys/kiac028). This Jiang lab paper tracked vacuole morphology through to GMC and guard cell stages, but did not characterize SLGCs or pavement cells. As the reviewer suggests, SLGCs that will differentiate into pavement cells will undergo a massive vacuole-driven expansion. Whether the change in vacuolar properties is visible in SLGCs and whether there are different morphologies for SLGCs that will divide or differentiate would be exciting avenues for future studies.

7) In the summary cartoon (Fig 6A) there is no feedback loop from number of neighboring signalling cells and cell size, yet when I was reading the text my impression was that this going to be an important element of the system (i.e., SLGC size influences the number of neighbours and number (and type) of neighbours influences SLGC size. Perhaps I have misunderstood, but it would seem reasonable for such a feedback system to be in place.

Thank you for asking this question, it has made us think about why we didn't put in such feedback and whether it would be appropriate. There is certainly work, particularly in the shoot apical meristem, showing that the number of neighbours can affect growth rates (and cell size) through changes in pressure (e.g. Long et al. 2020 Current Biology). Surprisingly, we don't detect significant relationships between SLGC growth rates and the total number of neighbours ($P = 0.34$) or the number of signaling neighbours ($P = 0.50$). As such, the primary influence of neighbours appears to be on SPCH degradation rates, rather than on growth rates.

Linear models ($N = 70$ cells)

- growth rate \sim total number of neighbours: $\beta = 0.082 \pm 0.086$, $t = 0.96$, $P = 0.34$
- growth rate \sim signaling neighbours: $\beta = 0.092 \pm 0.13$, $t = 0.68$, $P = 0.50$

It may also be useful to note the developmental origin of the SLGCs we track in our experiments. In leaf development the number of neighbours is determined by the history of division orientations + growth of a given cell and its neighbours. Plant cells have stiff walls and don't migrate, so for the most part an SLGC's neighbourhood appears to be determined by random (unpredictable) choices in division directions and ratio of daughter cell sizes (division asymmetry) made earlier. An example of SLGCs and their surrounding cells in WT is in Fig. 5a. SLGCs of a variety of sizes can have neighbours of different histories and lineages. Although larger SLGC cells have more neighbours, it is not because they make more neighbours.

8) In the references I couldn't see a reference to the work of. D'Ario et al (2021) DOI: [10.1126/science.abb4348](https://doi.org/10.1126/science.abb4348) which describes how the nuclear level of a KRP protein influences cell size control in plant stem cell niches. Is there a potential crossover here of the two systems

(SPCH and KRP)? Might be worth including in the discussion. Could this be the “SPCH-independent” required by their model? Or could they both be acting in the same pathway?

The D’Ario work is indeed foundational in the field, and we agree it should be mentioned in the discussion. We have added it now, with a consideration of why the molecules and mechanisms appropriate for the SAM are unlikely to be the same as for SLGC decisions (starting line 294). In short, the D’Ario paper suggests that small and large cells resulting from an asymmetric division inherit equal amounts of the G1/S phase inhibitor KRP4, and smaller cells must grow to dilute this inhibitor, thereby ensuring the SAM population maintains cell size homeostasis. SLGCs are not in a homeostatic growth phase, and the smaller SLGCs are more likely to divide than larger SLGCs, so dilution of a cell cycle inhibitor does not fit this situation.

Reviewer #2 (Remarks to the Author):

Review of the manuscript: “Multi-scale dynamics influence the division potential of stomatal lineage ground cells in Arabidopsis.”

The authors investigate the key question of how the critical decision—whether lineage precursors differentiate or continue dividing—is regulated, using stomatal lineage ground cell (SLGC) development in Arabidopsis. They employ various methods, such as long-term imaging and statistical modeling, to identify features that influence SLGC behavior at different spatiotemporal scales (including time, subcellular, and intercellular levels). Their findings emphasize the size of cells and SPCH concentration and cell-cell signaling, particularly from neighboring stem cells like meristemoids, in determining SLGC behavior. Interestingly, this decision-making process in SLGCs contrasts with that of meristemoids, as the authors' group previously discovered.

Given the challenges of studying the factors and transcriptomes involved in SLGC characteristics, as well as the limited available information, the discoveries and approaches used in this study—particularly the extensive imaging analyses and modeling—offer valuable insights to help bridge the knowledge gap in SLGC fate decisions during epidermal cell development.

I have a few comments on the manuscript.

It would be better to emphasize the importance of both cell size and SPCH concentration equally, as well as the existence of SPCH-independent pathways in SLGC fate determination. While the findings from the imaging analyses were sufficiently highlighted, the modeling results were underrepresented in the main figures. To address whether SPCH alone is sufficient to explain SLGC behavior, the authors used modeling. They discovered the existence of SPCH-independent pathways that link cell size and signaling to SLGC behavior, which is one of

the key highlights of their work. However, this is not mentioned in the abstract, and the detailed data are missing from the main figures.

We appreciate this viewpoint and have reorganized the text and figures. Some of the modeling results in Fig. S7 have been moved to a main figure (Fig. 7) and the previous Fig. 6 has been split into two parts with b-c as Fig. 6 and part a (the schematic summary) modified to become Fig. 8. We have added an explicit mention of the modeling to the abstract.

Figure 5B: The Y-axis title should be consistent across both graphs, and an explanation of the p-value is missing from the figure legend.

These data are actually different. The SPCH2-4A data are derived from a timecourse experiment (described in the Methods) where cells are imaged at 3-dpg and again at 5-dpg (rather than a timelapse experiment where cells are imaged every 40-45 minutes) so we do not have the appropriate time resolution to consider these cells to be measured "at birth".

What does this mean for our interpretation of the SPCH2-4A data? It could raise a concern that the SPCH2-4A cells are not actually dividing at larger cell sizes but that dividing cells *appear* to be larger simply because they are measured at a later stage of their cell cycles and have grown for longer. To address this concern we estimated how much larger cells would be just as a consequence of not sampling at birth. Since SLGCs grow ~1.5%/hour, we'd expect cells at the end of their cell cycles to be ~43% larger than if measured at birth (assuming a cell cycle length of 24 hours). Thus, in the very unlikely event that all of the SPCH2-4A cells were measured at the end of their cell cycles, we'd expect them to be at most 43% larger than their wild-type counterparts. If we look at the differentiating SLGC populations in Fig 5b we do see a slight increase in size, with the population of SPCH2-4A cells being 5.7% larger than wild-type cells (112.92 vs. 106.88 μm^2). When we look at the dividing cell populations in Fig 5b, however, we see that the SPCH2-4A cells are on average 83% larger than dividing wild-type cells (133.99 vs. 73.17 μm^2).

We therefore will leave the Y-axis labels as they are, and are confident that the conclusion that SPCH2-4A SLGCs can divide at larger sizes is valid.

We have added the *P*-value description to the figure legend: "*P*-values are from mixed-effects models with behaviour as a fixed effect and individual as a random effect."

Figure 6: It would make more sense for "(A) Model for SLGC division propensity" to be placed after (B) and (C).

Great suggestion—we have moved Figure 6a to Figure 8.

Lastly, the reference "Zeileis, A., Hothorn, T., 2002. Diagnostic Checking in Regression Relationships" appears to be missing from the text.

Thank you! We have removed it from the References section.

Reviewer #3 (Remarks to the Author):

The manuscript studies the fate decisions of stomatal lineage ground cells (SLGC) in Arabidopsis, specifically whether they differentiate or divide asymmetrically. The authors report that large cells tend to differentiate and propose a link between division behaviour and the transcription factor SPEECHLESS (SPCH) both at the neighbourhood scale (strength of cell-cell signalling) and at the subcellular scale (size of the nucleus and thus SPCH nuclear concentration). Using experimental and computational approaches, they propose specific decision pathways leading to differentiation or division based on cell size, number of signalling neighbours and SPCH levels.

The manuscript is well written with a good flow and a nicely presented story, with some room for improvement.

1. The main point to mention is the need for more explanations and details on the statistical and computational methods. For example, terms such as Gini impurity and cost complexity pruning may be unfamiliar to people outside the field (even for applied mathematicians working on modelling and model selection).

Thank you for the suggestion! We have made the following changes to the Methods (in red): “We first split our data into training and test sets (70:30) by randomly sampling without replacement. Next, we fit the CART algorithm to the training set, using Gini impurity as a measure of split quality. The Gini impurity was calculated as

$$G = \sum_{i=1}^C p(i) * (1 - p(i))$$

where C is the total number of classes (two in our case: division & differentiation) and $p(i)$ is the probability of selecting a datapoint in class i . The lower the Gini impurity, the more accurate the split. We then followed the "Post pruning decision trees with cost complexity pruning" protocol from *scikit-learn* to prune the decision tree to prevent overfitting (Pedregosa et al., 2011). We chose a cost complexity parameter value (`ccp_alpha`) that maximized testing accuracy.”

2. Some additional background on Arabidopsis development necessary to follow the manuscript would be useful to understand, for example, why 3 days post germination was chosen.

We have included additional text about the choice of time in lines 81-82 of the results: “This 3-dpg timepoint was chosen to maximize the number of active stomatal lineage cells for subsequent tracking”.

3. Line 227: it is not clear what is meant by randomly drawn asymmetry? Is it regarding the cell sizes of the daughter cells?

Yes, we defined the asymmetry of the mother cell divisions as $1 - \text{SLGC size}/\text{mother size}$. From our data, we obtained empirical distributions of this asymmetry parameter for each individual

plant, then fitted beta distributions to them (see Supplementary Table 2 for fitted parameters). The asymmetry of a given division in the simulation is drawn from the pertinent beta distribution. We recognize the wording in the text is confusing, and we have changed the text as follows.

Current text: The mother cells divide with a randomly drawn asymmetry...

Change: The mother cells divide asymmetrically with a randomly drawn asymmetry parameter... (lines 222-223)

4. The lineage decision tree diagram is helpful, but a flowchart with more detail and actually showing where each decision is made, how the degradation rate is modulated, and at what point the parameters are adjusted would be even more helpful.

We have modified Fig. S5 to provide a more detailed view of the model, and created a new main text figure (Fig. 7) that includes a brief summary of the model as well as the outputs.

We have added more details on how the degradation rate is modulated to the Methods:

“SPCH was degraded according to one of four degradation modes:

- Random (neighbour- and size-independent): decay constants (λ) were randomly drawn from an exponential distribution fitted to the empirical decay constant distribution of all cells in an individual
- Neighbour-based degradation: decay constants were randomly drawn from one of two exponential distributions, fitted to the empirical decay constant distributions of cells with 1 and 2+ neighbours, respectively (see Supplementary Fig. 5)
- Size-based degradation: per-square-micron decay constants ($\lambda/\text{cell size}$) were randomly drawn from an exponential distribution fitted to the empirical per-square-micron decay constant distribution
- Neighbour- and size-based degradation: per-square-micron decay constants were randomly drawn from one of two exponential distributions, fitted to the empirical per-square-micron decay constant distributions of cells with 1 and 2+ neighbours, respectively”

We have added the following to the legend of Fig. S5 to clarify when parameters are adjusted: “All distribution parameters and logistic model parameters were estimated from the data and fed into the decision tree models.”

5. Also, it is not clear at first glance whether the cells are actually simulated independently of each other, with no relationship between the 1000 mother cells and all properties drawn from specific distributions, or whether real space is also included? Again, better explanations with clearer details about methods and simulations are needed, and a flowchart would be very useful.

No space is included, and parameters and cell characteristics are independently drawn. We have remade Fig. S5 with more detail, and revised and moved parts of Fig. S7 into the main text to make the overall approach more obvious.

6. Regarding the AIC: depending on the sample size of the data, the corrected AIC (AICc) might be more accurate. In addition, it seems that the AIC values are quite similar and it is not easy to determine with certainty whether the top model is the best. In addition, Akaike weights could be calculated to see how likely each model is.

We assessed the models across the four replicates, so the total sample size is constant across models (262 cells). The models differ in the form of the logistic model describing the relationship between SLGC behaviour and different explanatory factors (i.e. the number of signaling neighbours, SPCH concentration, cell size, and their interaction terms). Since the correction term goes to zero with large sample sizes, we agree it is best to include it. The AIC scores reported in Supplementary Table 4 are now corrected AICs (AICc). The top model's AICc score is -903.265, and the second-best model has an AICc of -874.761, generally considered a large difference (28.504).

We have modified the Methods as follows (changes in red):

The **corrected** Akaike information criterion (AICc) was calculated for the total SSE across individuals using the following formula: $AICc = N \cdot \log(\sum SSE/N) + 2(k+1) + (2 \cdot k^2 + 2 \cdot k)/(N-k-1)$, where N is the total number of cells in the data (262) and k is the number of parameters in the **fate-determining logistic models, excluding the intercept**.

7. Are the data and code available?

The raw data for each figure are now in the source data file.

Code for the lineage decision tree model and the data used in the model are on [Figshare](https://doi.org/10.6084/m9.figshare.27931686v.1) (doi: 10.6084/m9.figshare.27931686v.1).

8. Supplementary Figure 4C is confusing. The probabilities of division are either close to 0 or close to 1 and a logistic regression is fitted to this. It is also very difficult to see the colours of the dots. What are we trying to show here?

Thank you for the feedback! To make the points more distinct, we have modified the colour scheme and jittered the points horizontally, in addition to vertically.

The probabilities of division are either 0 or 1 because the cells either divide or don't divide (they differentiate). The logistic regression shows that the total number of neighbours (both signaling and non-signaling; colour scale) is not predictive of SLGC behaviour after the number of signaling neighbours (x-axis) is accounted for. This suggests that it is the number of signaling neighbours, rather than the total number of neighbours, that is associated with SLGC behaviour.

9. Figure 2A-B: how is it established that the cell in A divides and that in B differentiates? Could you show this with images from a later time step?

Each cell was individually tracked in linked timecourses (Fig. 1) to confirm the relationship between SPCH behavior and cell outcome. It is an excellent idea to make this immediately clear by including a panel from the final timepoint and we have now done so in Fig. 2.

Minor comments:

10. The term “dip” is not clearly defined in the main text.

We have added this information to line 120 in the main text and have annotated the images in Fig. 2a-b to illustrate the dip.

11. On line 185: mention that total number of neighbours means signalling and non-signalling.

We have added this information to line 183 in the main text.